# Improved Corrosion Behavior and Biocompatibility of Porous Titanium Samples Coated with Bioactive Chitosan-Based Nanocomposites

**DOI:** 10.3390/ma14216322

**Published:** 2021-10-22

**Authors:** Cristina García-Cabezón, Vanda Godinho, Coral Salvo-Comino, Yadir Torres, Fernando Martín-Pedrosa

**Affiliations:** 1Departamento de Ciencia de Materiales e Ingeniería Metalúrgica, Escuela de Ingenierías Industriales, Universidad de Valladolid, Calle Paseo del Cace 59, 47011 Valladolid, Spain; fmp@eii.uva.es; 2Departamento de Ingeniería y Ciencia de los Materiales y del Transporte, Escuela Politécnica Superior, Calle Virgen de África 7, 41011 Sevilla, Spain; vfortio@us.es (V.G.); ytorres@us.es (Y.T.); 3Departamento de Química Inorgánica, Escuela de Ingenierías Industriales, Universidad de Valladolid, Calle Paseo del Cace 59, 47011 Valladolid, Spain; coraleugenia@hotmail.com

**Keywords:** porous titanium implants, electrodeposition, Chitosan/AgNPs, Chitosan/Hydroxyapatite coatings, corrosion characterization, in vitro behavior

## Abstract

Porous titanium implants can be a good solution to solve the stress shielding phenomenon. However, the presence of pores compromises mechanical and corrosion resistance. In this work, porous titanium samples obtained using a space-holder technique are coated with Chitosan, Chitosan/AgNPs and Chitosan/Hydroxyapatite using only one step and an economic electrodeposition method. The coatings’ topography, homogeneity and chemical composition were analyzed. A study of the effect of the porosity and type of coating on corrosion resistance and cellular behavior was carried out. The electrochemical studies reveal that porous samples show high current densities and an unstable oxide film; therefore, there is a need for surface treatments to improve corrosion resistance. The Chitosan coatings provide a significant improvement in the corrosion resistance, but the Chitosan/AgNPs and Chitosan/HA coatings showed the highest protection efficiency, especially for the more porous samples. Furthermore, these coatings have better adherence than the chitosan coatings, and the higher surface roughness obtained favors cell adhesion and proliferation. Finally, a combination of coating and porous substrate material with the best biomechanical balance and biofunctional behavior is proposed as a potential candidate for the replacement of small, damaged bone tissues.

## 1. Introduction

Bone tissue replacement is usually needed as a consequence of disease, trauma (accidents) or natural aging, and is a common solution to one of the most important public health problems. The implants rely not only on the quality and quantity of host tissue but also on the intrinsic properties of the implant materials, in addition to the bio-interface between the implant and the bone tissue (surface energy, biochemistry and topography) [1].

Among the different metallic materials employed for implants, Ti c.p. and Ti alloys are widely used as they combine high specific strength, low density, excellent corrosion resistance and good biocompatibility. Although these materials have a lower Young’s Modulus than other metallic materials, there is still an intolerable mismatch between their stiffness (100–110 GPa) and that of the cortical bone tissue (20–25 GPa). The stress-shielding that is generated can cause the resorption and fracture of the adjacent bone [2]. To overcome this issue, the use of porous materials has been proposed. A suitable pore content and size is essential to guarantee the adequate biomechanical properties (stiffness, mechanical and fatigue resistance) and the biofunctional balance (corrosion resistance and bone ingrowth) of the implant [3,4,5].

The space-holder (SH) technique has proven to be a promising methodology for the manufacture of porous Ti implants, since it is a route that allows pore sizes and distribution to be easily controlled [6,7,8,9,10]. On the other hand, although it has been previously reported that the walls of the pores obtained using the SH technique present a micro-scale roughness pattern that favors osteoblast adhesion [5,11], it is still necessary to implement superficial modification routes of the implants that improve osseointegration with the surrounding bone tissue. A detailed characterization of the porous substrates surfaces and the influence of the different porosities in balancing the mechanical properties and how it affects bacterial behavior can be found in our previous works [12,13,14,15].

Titanium implants have an excellent corrosion resistance that is attributed to the formation of a protective passive TiO_2_ layer. Nevertheless, the concentration of efforts associated with the introduction of pores could compromise this behavior. The contact with body fluids and the accumulation of reactive oxygen species at the implant surface can lead to the rupture of this passive layer, resulting in corrosion pits and the dissolution of Ti^2+^ ions in the patient’s body [16,17,18].

The use of chitosan-based nanostructured coatings such as Chitosan/AgNPs and Chitosan/Hydroxyapatite coatings could be a valid solution to solve both the following problems: improve the corrosion resistance of porous titanium implants and favor their connection with the bone. Chitosan is a bioactive natural polymer, biodegradable, non-toxic, with wound-healing activity and antibacterial properties used in a wide range of bio-engineering applications [19]. It has good adhesion to titanium substrates and contributes to obtain a surface with similar topography and chemical composition to an extracellular matrix [20]. In this work, we evaluate porous Ti substrates produced using the SH technique with large and rough pores covered by chitosan using electropolymerization. Moreover, the antibacterial properties of the polymeric matrix could be increased by the incorporation of AgNPs [21] and the bioactivity of the coating improves with the addition of hydroxyapatite particles. Recent studies evaluate the effects of Chitosan/Hydroxyapatite coatings deposited using electrophoretic methods on the osseointegration of Ti-based implants, including porous implants fabricated using selective laser sintering [22]. These investigations reveal an improvement on osteoblast proliferation and biocompatibility of the implants by in vitro and in vivo experiments. Nevertheless, the corrosion resistance of the coated implants was not evaluated. De Sousa and co-workers [18] investigated the corrosion behavior of bulk c.p.-Ti grade four after coating with hydroxyapatite and chitosan doped with silver nitrate.

The main objective of this work is to obtain a potential candidate for the replacement of small, damaged bone tissues. The selected coated porous material must have the best biomechanical (rigidity and mechanical resistance) and biofunctional (resistance to corrosion, improvement of osseointegration and avoid the proliferation of bacteria) balance. In this scenario, a detailed study of the electrochemical and biological behavior of chitosan-based coatings on Ti porous substrates produced using SH with different degrees of porosity is addressed. This study focuses on evaluating the influence of the pore content and the composition of the coatings on corrosion resistance and biocompatibility of the substrates.

## 2. Materials and Methods

Figure 1 shows a diagram that summarizes the manufacturing steps of the coated porous titanium substrates, as well as the tests carried out to evaluate the corrosion and cellular behavior of the manufactured materials.

### 2.1. Porous Ti Substrate Preparation and Characterization

In this work, porous Ti substrates with different degrees of porosity were successfully prepared using the space holder technique by using different amounts of ammonium bicarbonate, AB (30–60 vol.%). A spacer size range of 250–355 µm was selected to promote infiltration of coatings. Ti powders were obtained via hydrogenation/dehydrogenation, presenting irregular morphology and a particle size distribution of 9.7 μm (<10%), 23.3 μm (<50%) and 48.4 μm (<90%). These powders were mixed with the spacer particles using a Turbula T2C Shaker-Mixer (tmg machines, Birmingham, U.K.) for 40 min to ensure good homogenization, using a previously optimized protocol; details can be found in reference [6]. The green compacts were obtained by pressing at 800 MPa in a universal Instron 5505 testing machine (Instron, High Wycombe, UK). The spacers were thermally removed by heat treating the substrates in two stages of 12 h at 60 and 110 °C under a low vacuum condition of 10^−2^ mbar. A final step of sintering was carried out in a Carbolyte^®^ STF 15/75/450 ceramic furnace (CARBOLYTE, Derbyshire, UK) for 2 h at 1250 °C under high vacuum conditions (~10^−5^ mbar).

For comparison purposes, in this work, titanium substrates obtained using forging processes and using conventional powder metallurgical routes (1300 MPa, sintering at 1300 °C, for 2 h and high vacuum conditions) were also evaluated. Before performing the surface modification and characterization of porosity (Archimedes and image analysis), the surface of the samples was grinded and polished with magnesium oxide (MgO) and hydrogen peroxide (H_2_O_2_). The porosity fraction, size and morphology of the pores were preserved after preparation. Additionally, an estimation of the mechanical behavior, using empirical equations reported by the authors in previous works was performed [6]. Porous samples will be named after the amount of spacer used (30, 40, 50 and 60 vol.%).

### 2.2. Deposition and Characterization of Bioactive Coatings

Chitosan (CS) aqueous solution was prepared by dissolving 1.5 mg/mL chitosan powder (average MW = 45 kDa with a degree of acetylation >75.0%) into deionized water containing acetic acid 2% vol. and stirred for 24 h. AgNPs colloids were synthetized using the Creighton method [23] based on a reduction of 10^−3^ M AgNO_3_ with excess ice-cold 2 × 10^−3^ M NaBH_4_ solution. Chitosan-AgNPs (CS-AgNPs) coatings were prepared by the dissolution of 1.5 mg/mL of CS into a 50-milliliter AgNPs solution containing acetic acid 2% vol. Morphology and size of the AgNPs in the CS-AgNP films were studied using scanning electron microscopy (SEM). A QUANTA 200F SEM system (FEI, Hillsboro, OR, USA) was used to record the images of the samples; the average size of the AgNPs was about 30–40 nm. Hydroxyapatite (HA) particles were prepared by precipitation using a procedure described by Jarcho [24] and Pang [25] at 70 °C by the slow addition of a 0.6 M (NH_4_)_2_HPO_4_ solution into a 1.0 M Ca(NO_3_)_2_ solution, at 11 pH. The average size of the HA powders obtained from XRD diffractogram was found to be 50–75 nm, which is in agreement with the literature [24]. To prepare a Chitosan–Hydroxyapatite (CS/HA) composite solution, 0.025 g of chitosan powder was dissolved in 10 mL of a 0.2 M acetic acid solution under stirring conditions at 50 °C for 2 h and cooled at room temperature, then ethanol was added until 50 mL of solution was obtained. An amount of 0.15 g of HA powder was added into ethanol–water solution and stirred to obtain the homogeneous CS/HA composite solution.

All chemical reagents were purchased from Sigma Aldrich (Merck KGaA, Darmstadt, Germany). Aqueous solutions were prepared using Milli Q water (resistivity 18.2 MΩ∙cm).

These solutions were used as electrolytes to form the bioactive coatings using electropolymerization on the metallic substrates. The depositions were performed using an EG&G Parstat 273A potentiostat/galvanostat (Princeton Applied Research, Oak Ridge, TN, USA) at room temperature, with the classic three-electrode configuration: a platinum plate was used as a counter electrode, a Ag/AgCl electrode in a 3 M KCl solution as a reference electrode and the metallic substrates as a working electrode. All substrates were submitted to a final polishing procedure with a SiO_2_/H_2_O_2_ suspension, followed by cleaning in an ultrasonic bath with distilled water and ethanol. Before the electrodeposition, the solutions were ultrasonicated for 1 h to achieve a homogeneous dispersion of the nanoparticles.

CS and CS-AgNPs films were obtained using a chrono-amperometry technique at a constant potential of −1.5 V_Ag/AgCl_ for 900 s. For the CS-HA films deposition, chrono-potentiometry (CP) at a current density of 0.1 mA/cm^2^, for 1200 s, was applied. To avoid stability problems, all depositions were performed immediately after preparing the solutions and stirring was maintained during electrodeposition. The coatings were subjected to the peel test (ASTM D 3359), method A [26], to determine the adhesion of the coatings. To evaluate the morphology of the coatings and the homogeneity of the deposits onto the porous substrates, scanning electron microscopy was performed using a SEM-FEG FEI QUANTA 200F microscope (FEI, Hillsboro, OR, USA) equipped with Energy Dispersive X-ray Spectrometry (EDS). EDS measurements were performed to confirm the composition of the coatings. X-ray diffraction analysis was carried out using an Agilent SuperNova diffractometer (Agilent Technologies XRD Products, Yarnton, UK) using micro CuKα/MoKα radiation with CCD Atlas detector (Agilent Technologies XRD Products, OX5 1QU, Kidlington, UK) Atomic force microscopy scans were also performed to evaluate the surface roughness of the different coatings, 3D and 2D scans of (5 μm × 5 μm for CS and CS-AgNPs and 10 µm × 10 µm for CS-HA) were performed in a Cypher ES (Asylum Research, Santa Barbara, CA, USA) operated in the tapping mode.

### 2.3. Corrosion Behavior

To evaluate the effect of porosity on the corrosion resistance of the substrates and the protective character of the bioactive coatings, electrochemical methods were employed. Open circuit potential (OCP), anodic polarization (PA) measurement and electrochemical impedance spectroscopy (EIS) were performed using a potentiostat/galvanostat (PARSTAT 273A, Princeton Applied Research, Princeton, NJ, USA) and an impedance analyzer (Solartron SI 1260, Solartron Analytical, Farnborough, UK). The tests were carried out in PBS physiological solution at 37 °C, in a conventional three-electrode cell, with a graphite rod counter electrode and a saturated calomel electrode (SCE) as a reference electrode. The OCP was recorded for 3600 s of immersion of the samples in the electrolyte solution. Potentiodynamic anodic potential curves of sintered samples were performed according to ASTM G-5 standard [27] prior conditioning and potential stabilizing. The initial potential was 250 mV below V_OCP_, the final potential was 1000 mV_SCE_ and the potential scan rate was 50 mV/min. The corrosion parameters, including corrosion potential (E_corr_) and corrosion current density (I_corr_), were obtained from polarization curves using Tafel’s analysis and polarization resistance was calculated using Stern–Geary equation. EIS measurement was performed at open circuit potential from 10 MHz to 0.1 Hz at 10 data cycles/decade after measurement of the free potential from the immersion time during 1800 s. All electrochemical experiments were repeated three times to verify reproducibility.

### 2.4. In Vitro Study

In vitro experiments were performed to evaluate the biocompatibility of the samples comparing the bare titanium substrates with the coated substrates. MC3T3-E1 mouse preosteoblasts (CRL-2593™, ATCC^®^, Manassas, Virginia City, NY, USA) were cultured in α-MEM with nucleosides without ascorbic acid (Gibco, Brooklyn, NY, USA) with 10% fetal bovine serum (Gibco Brooklyn, NY, USA), at 37 °C and 5% CO_2_.

The cells were trypsinized and seeded on 24-well plates at a density of 10^4^ cells/cm^2^. These were left unaltered for attachment in complete growth medium at 37 °C and 5% CO_2_ for 2 h. Then, the Ti substrates with and without the bioactive coatings were placed directly in contact with the cells, as specified in the ISO 10993-5 [28]. Cell viability (LIVE/DEAD assay; Invitrogen, Carlsbad, CA, USA) and metabolic activity (alamarBlue assay; Invitrogen, Carlsbad, CA, USA) were evaluated after 1 and 3 days. The samples were removed from wells and a Nikon Eclipse Ti-E fluorescence microscope coupled to a Nikon DS-2MBWc digital camera (Nikon Corporation, Tokyo, Japan) using the NIS-Elements Advanced Research software (version 4.5; Nikon Corporation, Tokyo, Japan) was used to take images of living and dead cells. All the experiments were performed in triplicate, and it allowed the automatic imaging of almost the full well surface for quantitative determination of cell viability. The metabolic activity was measured in a fluorescence plate reader.

## 3. Results and Discussion

In this section, a first detailed study on the morphology of the coatings is presented to evaluate the quality of the biopolymeric coatings on the porous substrates’ surfaces. (Details on the characterization of the porous substrates can be found in references [12,13,14,15]). Next, the influence of the pore content of titanium substrates and the chemical composition of the chitosan-based nanocomposite coatings on the corrosion resistance and cellular behavior (adhesion and proliferation of osteoblasts) are discussed.

### 3.1. Microstructural Characterization of the Coatings

Figure 2a–c show the SEM micrographs of representative coatings deposited on the porous Ti substrate prepared with 30% vol of spacers: CS/30 vol.%, CS-AgNPs/30 vol.% and CS- HA/30 vol.%, respectively. All the coatings are conformal to the porous substrates and present a smooth surface. Figure 2a demonstrates the presence of a coherent, crack free and CS-coated layer completely covering all the macro-pores of the sample. A smooth, uniform and folded layer of a CS-AgNPs film covering the matrix and pores can be observed in Figure 2b. Higher magnification SEM micrographs exhibit spherical nanoparticles of AgNPs, 30–40 nm, which are densely backed together with a CS matrix, as shown in Figure 2d; the EDS spectra show the presence of C and N, the required constituents of CS, and Ag in addition to the peaks of the titanium substrate. This confirms that AgNPs were incorporated with a high percentage inside the CS matrix. Figure 2c shows the surface of the CS-HA coatings, the deposits are relatively dense and contain HA particles and some HA agglomerates of small size, distributed in a CS matrix that can be observed in more detail in Figure 2f with higher magnification. Figure 2e,g show cross sections of CS-AgNPs and CS-HA coatings. The average thicknesses of the CH, CH-AgNPs and CH-HA coatings were 3, 3.5 and 5.5 μm, respectively, the thickness increasing slightly with the addition of AgNPs and more noticeably with HA incorporation. EDS data confirm the incorporation of HA into the coating, and Ca, P and O peaks were observed in addition to those of C and N from the chitosan and Ti from the substrate. In all cases, the electrodeposited biopolymeric coating covers the pores’ surface forming a protective film on the porous titanium surface that could prevent the entry of aggressive ions and, thus, could increase the corrosion resistance of the porous implants.

Figure 3 presents representative results of XRD phase analysis for the coated samples, which featured a superposition of sharp peaks corresponding to highly crystalline Ti substrates, with broad peaks corresponding to chitosan. Semi-crystalline chitosan (broad peaks at 2θ = 10° and 2θ = 20°), which has been con-firmed by comparison with a Joint Committee on Powder Diffraction Standards (JCPDS card, file No.039-1894). For the CS-AgNPs coating, silver peaks were identified (JCPDS card, file No. 04-0783), whereas for the CS-HA coating, hydroxyapatite peaks (JCPDS card, file No. 09-0432) were observed. In addition, α-HCP titanium peaks were visible, substrate (JCPDS card, file No. 44-1294), especially for the CS coatings. The XRD pattern indicated that chitosan displays a more amorphous form in all coatings, which may have biomedical applications [19]. These results confirmed the incorporation of nanoparticles in the CS-AgNPs and CS-HA coatings, which could further enhance their biocompatibility.

Figure 4 shows representative AFM surface topography images of the CS/40 vol.%, CS-AgNPs/40 vol.% and CS-HA/40 vol.% samples. The results are presented in 2D and 3D images. Surface roughness of these coatings through Sa (arithmetical mean deviation) and Sq (root mean square height) parameters were calculated from these images. As it could be expected, the CS coating surface, Figure 4a, is very smooth with very low roughness values Sa = 1.42 and Sq = 1.65 nm. The homogeneity of the coating surface indicates that the pores have been completely covered.

After the incorporation of AgNPs on the CS coating, the roughness of the sample was increased, Sa = 11.12 nm and Sq = 13.42 nm were obtained. The changes on the topography can be clearly seen in the images of Figure 4b. The AgNPs incorporation produced a surface with heterogeneous heights and a grainy texture. It is observed that the nanoparticles are spherical and monodispersed, without cavities and there is no indication of aggregation. This is due to the fact that some capping reagents present in the CS seem to act as ligands that effectively stabilize the synthesized AgNPs [29]. The size of the nanoparticles observed using AFM is clearly higher than those measured using SEM, probably due to the formation of the CS shell around the nanoparticles.

Figure 4c illustrates the surface properties of CS-HA composites in 3D and 2D topographies using AFM. The figures exhibit dispersed HA particles in the polymer matrix. As can be seen, the HA particles are surrounded and embedded into the polymer. It seems that the penetration of HA particles into the chitosan matrix leads to the formation of aggregates and agglomerates, and it can explain the increase in the recorded roughness. The presence of these aggregates has been already referenced in the literature [30] and it could be due to the high specific surface energy of HA particles. As in the case of AgNPs, we can see that the HA particles are covered with a continuous smooth CS layer and, therefore, the particle sizes are higher due to the CS layer. The aggregates of HA generated during the electrodeposition appear as high areas with lighter colors than their surroundings. The roughness of these areas was also measured, the resultant values were Sa = 57.16 nm and Sq = 93.86 nm for the aggregates, and Sa = 23.62 nm and Sq = 26.72 nm for the isolated particles. Moreover, it has been reported that the adhesion and proliferation of cells increased on nano/micrometer scaled surfaces, as shown in CS-AgNp and CS-HA coatings [31]; therefore, the incorporation of particles into the CS matrix could improve biocompatibility (see details in the Section 3.3).

### 3.2. Corrosion Behavior: Influence of Porosity and Type of Coating

The influence of the porosity of samples and applied surface modifications on the corrosion behavior of wrought and PM samples was evaluated using the open circuit test, the anodic polarization curves and EIS measurements.

The OCP evolution during the immersion period for the fully dense (wrought) and PM unmodified titanium samples is shown in Figure 5a. For the first few minutes of immersion, the OCP values shifted slightly, but quickly reached steady state conditions for all the samples. It can be seen that the porous samples have a more anodic value compared to the dense ones, indicating their higher thermodynamic stability. The open circuit potential values for 50 and 60 vol.% showed more positive values, which could be attributed to the formation of a more stable and protective layer inside the pores, acting as a barrier to prevent the release of metal ions.

On the other hand, the open circuit potentials of CS-, CS-AgNPs- and CS-HA-coated samples were studied with immersion time in PBS solution at 37 °C. As an example, Figure 5b shows the variation of open circuit potential with time for uncoated and coated 40 vol.% samples. It was found that all the studied coatings cause an important ennobling in OCP values, which has been observed in all cases, for dense and porous samples.

Figure 6 shows final OCP values for dense and porous samples with and without bioactive coatings. In all cases, the uncoated samples are the lowest protected ones, presenting the most negative OCP values. CS samples exhibited more positive values than the uncoated samples and the CS-AgNPs nanocomposite coating provided the best performance for fully dense and 0, 30 and 40 vol.% samples. The beneficial effect of mixing nanoparticles and CS has been also observed for fully dense Ti [32] and a 3D-Printed Porous Titanium Alloy Scaffold [33] showing a positive OCP value that demonstrates a higher stability and spontaneity than CS coating, showing a much stronger coating for porous samples. The pores could improve the adherence of coatings and, therefore, improve the enhancement of corrosion resistance [14,34,35].

Figure 7a shows the representative potentiodynamic polarization curves for the uncoated samples. In the anodic polarization scan of porous samples, although a passivation plateau was not observed as stable, as in the fully dense sample (wrought), the current density gradually increased with small fluctuations and finally exhibited a passivation region. These differences could be attributed to the heterogeneities of the passive film formed in the innermost pores. Furthermore, the porous samples showed a higher corrosion density compared to the dense samples, which is in agreement with the literature [36,37,38]. In this regard, it should be noted that it is not possible to accurately measure the actual wet surface area of porous materials and the geometrical surface area used to determine the current density values. With increasing the substrates porosity, the size of the real exposed surface increases, but the current density registered does not decrease because it was defined for the geometric surface. Additionally, the deviation of the measurements increases with increasing porosity. Therefore, an increase in the current density and corrosion rate with the level of porosity is to be expected. Furthermore, it is well known that the morphology of pores, whether isolated or interconnected, plays an important role in the electrochemical behavior of porous materials [39]. In the case of isolated pores, the electrolyte could be trapped, resulting in crevice corrosion; however, the electrolyte could flow freely in interconnected pores, allowing oxygen substitution and the formation of passive layers [39]. In this work, no crevice corrosion has been observed and the higher current density of the 40 and 60 vol.% samples with larger pores is a result of the larger surface area exposed to the solution.

Table 1 shows the results of E_corr_ and i_corr_ measured using the Taffel method and R_p_ measured using the Stern–Geary method. In good correlation with the OCP values, the E_corr_ of the more porous samples has a more anodic value compared to the dense samples, representing an oxide film with a higher thermodynamic stability. This shift of the anodic curves toward higher potentials with increasing porosity has been reported by other authors who studied the influence of pores on the electrochemical behavior of Ti [40,41] and Ti alloys [37,42]. However, Shivaran et al. [43] observed that the corrosion potential of Ti alloys decreases with increasing porosity, contrary to what has been seen in the previous research works.

Figure 7b shows polarization curves of uncoated and coated 50 vol.% samples, it can be observed that E_corr_ values of the coated samples are more anodic than the porous substrate, especially when CS was electrodeposited with AgNPs or with HA. These results clearly support the previously obtained OCP results. It is observed that after coating both cathodic and anodic current densities are significantly reduced in a large potential range, especially for the CS-HA biocomposite coating that showed a more stable and larger passive region than the others.

Similar behavior was observed for the dense and for the rest of the porous samples; in all cases, the presence of the coating shifts the polarization curve to lower current densities and more positive potentials, indicating higher corrosion resistance. The polarization parameters, listed in Table 2, show that the protection efficiencies of the three coatings are different in function of the porosity level. For coated samples, the protection efficiencies (PE%) are also given in Table 2 and were calculated from the following equation:PE% = i_uncoated_ − i_coated_/i_uncoated_(1)
where i_uncoated_ and i_coated_ are the uncoated and coated corrosion current densities, respectively.

When CS was electrodeposited, a slight beneficial effect was observed on a fully dense sample. A similar effect was reported in the literature [32,44] for non-porous titanium alloys; moreover, in some works this protection was not observed [45,46]. The incorporation of metallic NPs [47], PVP [46], HA [29,48] in the chitosan coating on dense titanium alloys, results in a significant decrease in the current density and an increase in the corrosion potential. What is noteworthy is that the incorporation of AgNPs into the chitosan coating resulted in a significant increase in PE% for fully dense and 0 vol.%, while for porous samples it was found that PE% for CS-HA is the highest, reaching 96.9%. Therefore, the presented results show that the beneficial influence of the incorporation of NPs or HA into CS coating for corrosion resistance also appears on porous titanium. To explain this, it must be considered that CS coatings are porous and with little thickness; therefore, the deposition of NPs or HA decrease the number of pores in the CS coatings. The increase in size and roughness inhibits the transfer of the aggressive solution through the coating to the surface of the porous titanium.

EIS measurements were performed to further investigate the influence of porosity on the corrosion behavior of dense and porous samples. The impedance data are presented as Nyquist and Bode plots (impedance modulus versus frequency and phase angle versus frequency) in Figure 8. The incomplete and depressed semicircle of larger radius corresponds to the fully dense sample in Figure 8a, while the incomplete semicircles of the smaller radius correspond to the porous samples. The most porous sample, 60 vol.%, presented the smallest radius semicircle. In addition, the impedance modulus in the mid and low frequency range, observed in the Bode diagram, Figure 8b, was lower for the most porous samples and the phase angle moved away from −90°, which is the atypical capacitive behavior of a compact oxide film, when the porosity increased, Figure 8c. All these data suggest a higher susceptibility to corrosion in the porous samples, which can probably be attributed to the heterogeneities of the oxide film formed on the inner surface, due to the difficulties of the electrolyte to access the inner of the pores.

The equivalent circuit of Figure 8d is used to interpret EIS data, it is the most commonly used for titanium in physiological electrolytes [49]. The high frequency in the proposed circuit is related to the electrolyte resistance (R_S_), and the responses at low and medium frequencies are related to the resistances of the inner barrier layer (R_in_) and resistance of the porous outer layer (R_out_) of TiO_2_, respectively, in the case of fully dense samples. In the case of porous samples, Rout corresponds to the electrolyte resistance inside the pores [50] and R_in_ to the resistance of the inner passive layer.

The depression of the semicircle in the Nyquist diagram and the phase shift (>−90°) suggest the use of a constant phase element (CPE) instead of a pure capacitor. The impedance of a CPE is defined as follows:(2)$$Z_{CPE}=\frac{1}{C{\left(j\;\omega \right)}^{n}}\;$$
where *CPE* for n = 1 corresponds to an ideal capacitor; for n = 0, CPE is an ideal resistance; and for n = −1 *CPE*, t is a pure inductance. *CPE*_out_ corresponds to the outer surface capacitance and *CPE*_in_ to the inner layer capacitance. 

Table 3 shows the fitting parameters. It was observed that there was not much variation in the values of R_s_, the obtained values of CPE_in_-n were close to 1, which is indicative of a quasi-capacitive behavior of the formed passive layer. The Rout values of the porous samples were lower than the R_in_ values, indicating the formation of a weaker protective layer inside the pores. The Q_in_ and Q_out_ increased, and the resistances R_in_ and R_out_ decreased with the increasing porosity. The higher capacitances and lower resistance values on the porous samples compared to the dense ones indicated a poor quality of the passive film and the formation of an instable oxide film in a passive state. Similar results have been found in other studies [43] to point out that the higher exposed surface area and the difficult electrolyte penetration to get into the pores leads to an instable, heterogeneous and insufficient thickness of oxide film. Other authors [51] have reported that the corrosion behavior of porous samples is a function of the pore characteristics. Open and interconnected porosity of porous samples provides access to more electrolytes, which greatly fail to maintain the protective titanium oxide layer on the alloy surface. In this sense, it can be observed that the sample obtained with a 60 vol.% spacer does not follow the improvement trend experienced by the 50 vol.%. This fact could be related to the “worse sinterability” of the porous samples (60 vol.%), the “weakness” of the necks that are formed between the titanium powder particles that, in this case, could be preferential sites of corrosion.

In order to fully understand the influence of the coatings on the corrosion processes of dense and porous titanium, electrochemical impedance spectroscopy for the three coatings were measured and the relevant corrosion parameters by fitting the EIS results to the circuit given in Figure 8e are presented in Table 4. For coated samples, an additional time constant has been added; R_coat_ and CPE_coat_ are the resistance and capacitance of coated film, respectively. The protection efficiency for coatings was calculated and tabulated in Table 4 using the following equation:PE% = R_coat_ − R_uncoated_/R_coated_ × 100(3)
where R_coat_ and R_uncoated_ are the total resistances for uncoated and coated samples, respectively.

It was observed that for dense and porous samples, the impedance values for the coated samples were higher than those for the uncoated samples. After the incorporation of AgNPs or HA into the CS matrix, in general, high impedance values were obtained from medium to low frequencies, suggesting high corrosion resistance. This behavior suggests that the deposited biocomposite film acts as a protective layer against ion diffusion and corrosion. This is in good agreement with the results of potentiodynamic measurements.

According to Table 4, the barrier inner layer is the main mechanism of corrosion protection for coated porous and non-porous samples. Therefore, the innermost layer resistance, R_in_ was higher than the resistance of R_coat_ and R_out_. Generally, the fitting parameters presented higher R_in_ values following the sequence CS-HA > CS-AgNPs > CS >substrate. This indicates the improvement in corrosion resistance after using the biocomposite coatings.

The efficiency of coatings is a function of the degree of porosity of the substrate. The electrodeposition of CS coating increased R_in_, especially for porous samples; therefore, the pores could increase the adherence of CS coating. The incorporation of AgNPs also significantly increased the PE% for the samples with higher porosity (30 and 40 vol.%) and the incorporation of HA was especially effective for 50 vol.% (PE% = 93.5). The PE% value of 60 vol.% was clearly the lowest for the CS-HA coating, indicating that the high degree of porosity makes it difficult to obtain a continuous, homogeneous and adherent corrosion-resistant coating.

Figure 9a,b show a Bode plot for the CS-HA samples; it was observed that once the coating was applied, the porous samples have a similar or even lower impedance modulus and more negative phase shift angle than the dense sample, except for the sample 60 vol.%, which showed the lowest impedance. On the contrary, the coated 50 vol.% sample showed the best behavior, which agrees with the anodic polarization behavior, Figure 9c. In this context, CS-HA coatings have been successfully developed for the protection of non-porous 316 L SS [31,52]. The use of HA in CS polymer matrix enabled the formation of a dense, adherent and thick film with higher impedance values that can be attributed to the increased water resistance of the polymer matrix due the incorporation of HA particles [53]. The results presented above confirm the role of these coatings in improving the corrosion resistance of the porous substrates that is crucial for a good candidate for implants. The SEM micrographs (not shown) of the cross sections of the coatings after the corrosion tests confirm this good performance, as the thicknesses of the coatings were similar to those recorded before the tests. The next step is evaluating the biocompatibility of these materials. The following section of this work addresses the influence of substrates porosity and the nature of the biopolymeric coatings on the adhesion and proliferation of osteoblasts.

### 3.3. In Vitro Cells Studies: Influence of Porosity and Type of Coating

Uncoated titanium samples with 0, 30 and 50 vol.% spacers were tested to assess how porosity can affect biological properties. After titanium coated with CS, CS-AgNPs and CS-HA were tested to check the effect of the coatings of fully dense and porous samples in vitro cells behavior. A LIVE/DEAD assay was tested on seeded MC3T3-E1 preosteoblast cells in contact with the conventional PM and porous metallic substrates, compiling the results in Figure 10a–c. Viable and fixation-phase osteoblasts were observed with high percentages of cell viability, indicating the biocompatibility of the three substrates. However, some differences were observed between the almost dense and the porous substrates. In the case of porous samples, the number of cells increased and they became bigger (Figure 10b,c). This indicates that the porosity achieved a better cell response as compared to the conventional PM denser sample. The difference between the 30 vol.% sample and the 50 vol.% sample were negligible.

The differences between day one and three confirmed that cells proliferate over time on all three substrates. Figure 10d–f show MC3T3-E1 cells were capable of proliferating over the flat surface and inside the pores, increasing the cell density. In fact, on day three, the cells covered the entire porous structure of the 30 vol.% sample, filling the pores completely. However, some pores in the samples that were prepared with a higher percentage of spacer volume (50 vol.%) were not completely covered and a heterogeneous distribution was observed at the bottom of Figure 10f. The most porous samples need a longer period of time to achieve complete and homogeneous cell surface coverage [13].

Several authors have also reported that porosity has a strong influence on the cell viability, adhesion, proliferation and differentiation of an osteoblast [54,55]. The increased roughness values inside the pores and the increased surface contact area of porous samples may promote a higher density of osteoblast cells compared to the dense ones [56]. The results obtained in this work are, to some extent, in accordance with observations already reported [57,58].

In the literature, it has been reported that the presence of chitosan and modified chitosan coatings has a beneficial effect on the biocompatibility of dense titanium [33,47]. The aim of this study was to evaluate the effect of chitosan coatings on the cytocompatibility of porous titanium samples manufactured using the space holder technique compared to cell adhesion and proliferation of MC3T3cells over practically dense coated samples.

Figure 10g–x show fluorescence images of the morphology of an MC3T3-E1 cell cultured on the CS, CS-AgNPs and CS-HA samples at day one and the respective images after 3 days of culture. On day one, the cell viability on the samples containing chitosan was similar for the dense and porous samples, but the CS coating provides a more notable increase in cell density for the dense sample than those grown on the uncoated sample. In the case of CS-AgNPs- and CS-HA-coated samples, the number of cells clearly increased for the three substrates as compared to the non-coated samples. This suggests that the incorporation of AgNPs or HA into a CS coating achieved a better cell attachment. On day three, cells grown on CS-AgNPs and CS-HA spread better and covered almost all the available surface area. No significant differences were found between the dense and porous samples.

AlamarBlue reduction results showed the metabolic activity of MC3T3E1 cells at 1 and 3 days of cell incubation; these are presented in Figure 11a,b, respectively. A remarkable increase in metabolic activity with time that is associated to an increase in cell proliferation was detected. Cell viability was expressed as a percentage of the uncoated conventional PM samples (control). An increased cell viability of osteoblast growth was observed on the two uncoated porous structures compared to the control surface at day one. Both the time and type of substrate showed significant differences at the 0.05 level. At 3 days, the 50 vol.% samples achieved lower levels of cell metabolic activity than the 30 vol.% samples; it has been previously reported that higher interconnected porosity leads to cells settling to the bottom of the culture plate and fewer cells being able to adhere to the sample [15].

The presence of the CS coating increased the cell viability on the practically dense sample after 1 day, while at 3 days, it decreased and did not show statistical differences and all the surfaces behaved similarly. For both times and for all the substrates, the coated samples showed an increase in cell proliferation with AgNPs or HA incorporation into polymer matrix. The negative effect of CS on cell viability has already been observed in fully dense titanium alloys [59] and has been attributed to the high water absorption of CS leading to the formation of soft gel-like structures that prevent the proper adsorption of cell adhesive proteins from the culture medium [60]. It is known that surface roughness and topography at the micro- and nanoscale influences cell morphology and proliferation [61]. Therefore, the improved cellular response of the composite coatings can be related to the increased surface roughness of biocomposites that has been observed using AFM.

On the other hand, it is known that HA can enhance protein adsorption through the release of Ca^2+^ ions [46] and can increase the stiffness of CS., e.g., hydrogen bridge bonds can form between the amino groups of HA and the hydroxyl groups of CS, which favors cells proliferation and propagation [62]. In addition, the HA coating can protect the metallic substrate from corrosion providing alkaline conditions suitable for cell growth [46]. The results obtained in this work indicate that biocomposites of HA and biodegradable polymers such as chitosan also improve biocompatibility in porous structures.

## 4. Conclusions

CS and modified CS coatings with AgNPs and HA obtained by electrodeposition were successfully deposited on all the titanium substrates. This non-expensive coating technique proved to be ideal to be used for porous Ti substrates candidates for small implants.

As expected, potentiodynamic polarization tests showed that the porous samples did not present a stable passivation plateau and exhibited higher corrosion current values than dense samples; the current density increased by almost two orders of magnitude in the most porous samples. EIS studies also revealed that the porous samples presented a less stable oxide film leading to lower protection properties. The complex geometry and limited interconnectivity of the pores could hinder the penetration of the electrolyte through the inner pores, resulting in a less protective oxide film formed in the pores’ surface. Consequently, the possible occurrence of a local corrosion attack should be considered when using porous implant materials and thus the need for surface treatments to improve corrosion resistance.

The results of the electrochemical studies showed that coatings provide corrosion protection of titanium substrates, especially for porous samples; thus, the porosity of the substrates improves material adherence. The incorporation of AgNPs and HA particles to the CS coating increased the open circuit potential, polarization resistance and impedance values, which is attributed to the enhancement of the protective properties of the passive layer by the incorporation of highly dispersed AgNPs or HA particles. Corrosion studies showed that the CS-HA coating shows the best corrosion performance, the maximum PE% was obtained for the CS-HA coated 50 vol.% sample.

In vitro tests showed an enhanced cell adhesion, spreading and proliferation of osteoblasts for porous samples compared to those that are practically dense. However, a higher interconnected porosity could have a negative influence in the cell proliferation of an osteoblast. In addition, preosteoblast cells seeded on the presence of the surfaces covered with coatings were highly viable results, even showing an increase in cell number over time as a consequence of cell proliferation, especially for composites with AgNPs or HA.

The combination of the CS-HA biocomposite film on the 50 vol.% porous titanium substrate has demonstrated an excellent performance in corrosion resistance and biocompatibility; this is the substrate-coating combination presenting the best biomechanical and biofunctional balance, being a promising alternative material for small implants to be used in biomedical applications.

## Figures and Tables

**Figure 1 materials-14-06322-f001:**
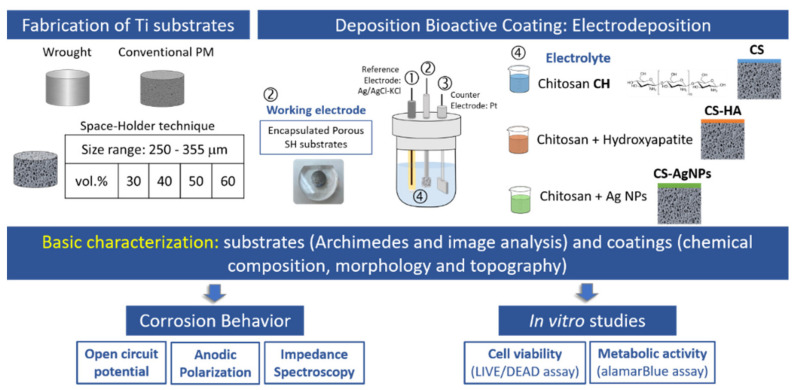
Manufacturing steps of coated porous Ti substrates and its characterization.

**Figure 2 materials-14-06322-f002:**
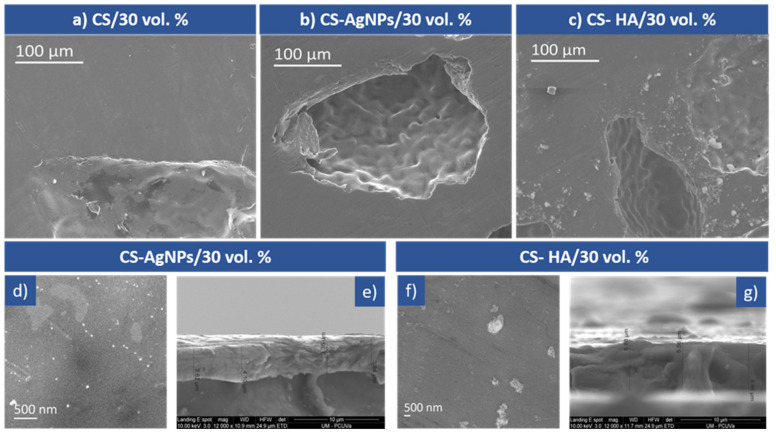
SEM micrographs of coated 30 vol.% samples: (**a**) CS, (**b**) CS-AgNPs, (**c**) CS-HA coatings, at low magnification, (**d**) higher magnification detail of CS-AgNPs, (**e**) cross section of CS-AgNPs/30 vol.%, (**f**) higher magnification detail of CS-HA, (**g**) cross section of CS-HA/30 vol.%.

**Figure 3 materials-14-06322-f003:**
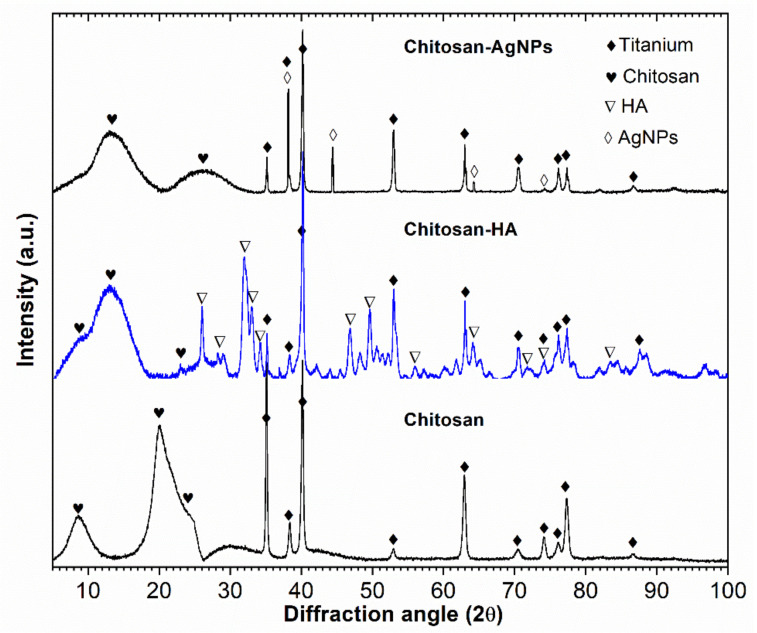
Results of XRD representative analysis of the 30 vol.% coated samples.

**Figure 4 materials-14-06322-f004:**
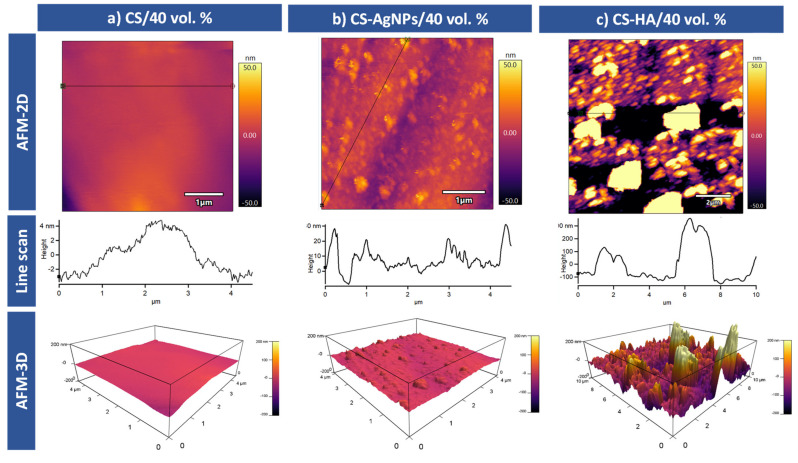
AFM topographic images for coated 40 vol.% samples: (**a**) CS, (**b**) CS-AgNPs and (**c**) CS-HA coatings.

**Figure 5 materials-14-06322-f005:**
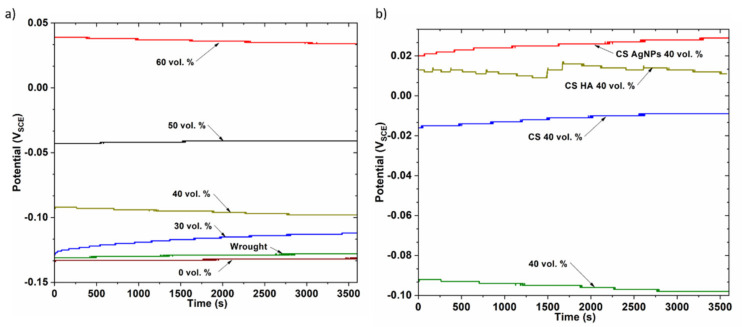
Evolution of OCP during immersion time: (**a**) effect of porosity, (**b**) effect of the coating for 40 vol.% sample.

**Figure 6 materials-14-06322-f006:**
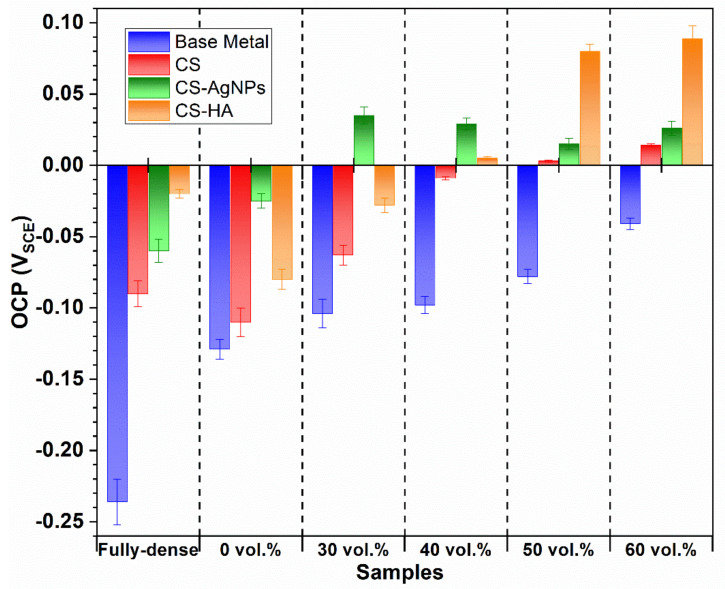
OCP final values of the immersion tests for uncoated and coated samples.

**Figure 7 materials-14-06322-f007:**
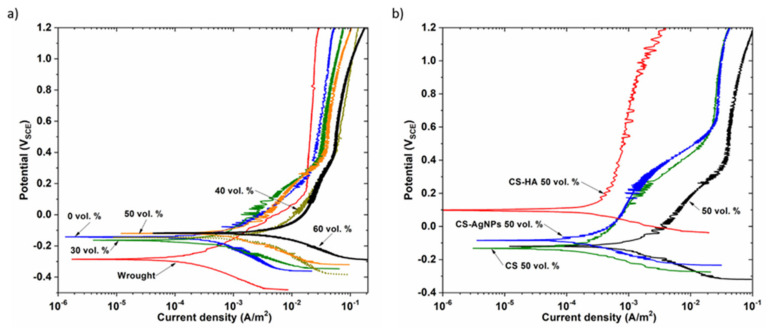
Potentiodynamic polarization curves for the samples: (**a**) uncoated and (**b**) 50 vol.% sample coated.

**Figure 8 materials-14-06322-f008:**
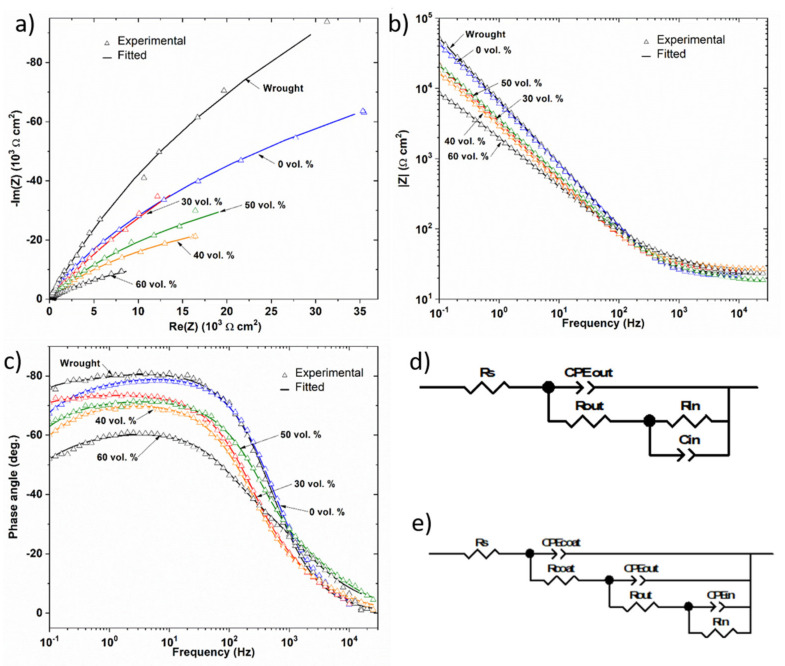
Effect of porosity on EIS depicted in (**a**) Nyquist and (**b**) and (**c**) Bode plots of uncoated samples. Results of fitting to equivalent electric circuit are included. Equivalent circuit proposed for the fitting of EIS spectra for: (**d**) uncoated and (**e**) coated samples.

**Figure 9 materials-14-06322-f009:**
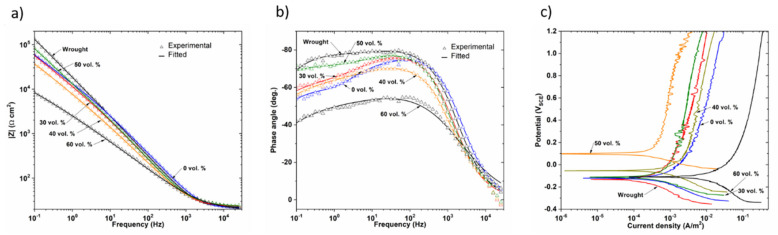
Effect of porosity of the substrate for CS-HA coated samples (**a**) and (**b**) Bode plots (**c**) potentiodynamic polarization curves.

**Figure 10 materials-14-06322-f010:**
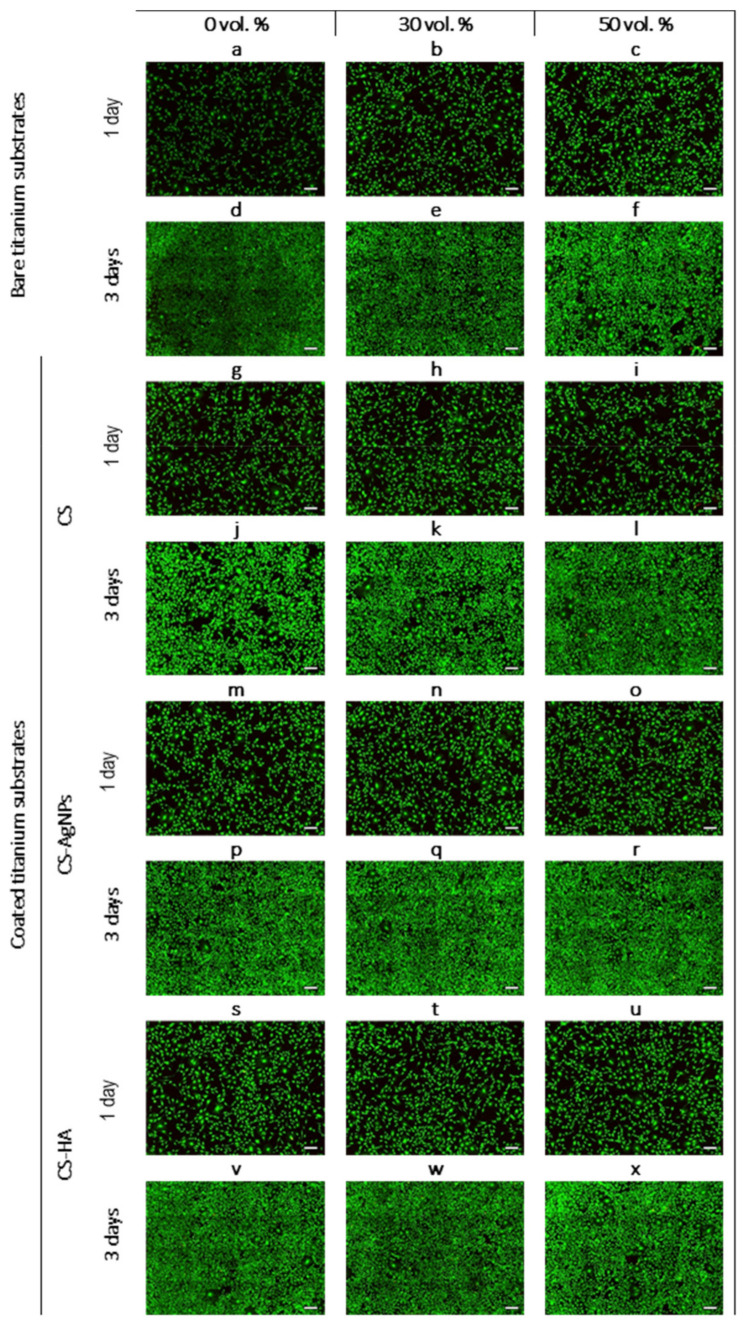
Fluorescence images of the morphology of MC3T3-E1 cell cultured on bare titanium substrates (**a**–**f**) and titanium coated with CS (**g**–**l**), CS-AgNPs (**m**–**r**) and CS-HA (**s**–**x**), during 1 and 3 days. Scale bar for mosaic images: 50 µm.

**Figure 11 materials-14-06322-f011:**
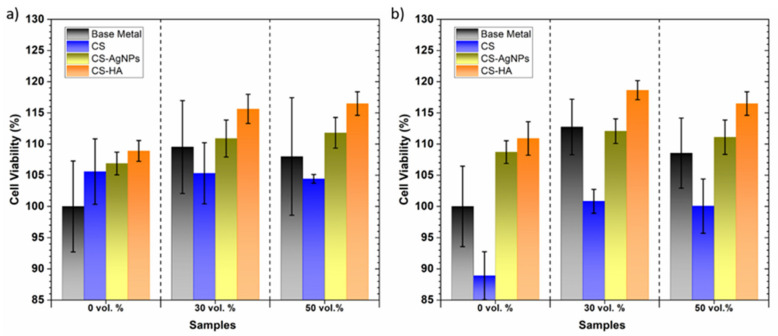
Cell viability study of coated and uncoated titanium substrates. (**a**) 1 day and (**b**) 3 days. Surfaces normalized to the positive control (uncoated samples, considered as 100% metabolic activity).

**Table 1 materials-14-06322-t001:** Values of Ecorr and icorr measured using the Taffel method and RP measured using the Stern–Geary method.

Sample		E_corr_ (mV)	β		I_corr_ (A/cm^2^) 10^−9^	R_p_ (µΩ/cm^2^)
			Anodic	Catodic		
Wrought titanium	Fully dense	−285	143.7	117.8	14.5	160
PM conventional	0 vol.%	−142	168.5	220.1	65.0	63.5
Space-holder technique	30 vol.%	−164	254.8	139.2	61.7	79.7
	40 vol.%	−146	799.0	572.9	1330	12.0
	50 vol.%	−119	629.7	265.1	320	26.1
	60 vol.%	−118	656.86	209.9	1260	58.5

**Table 2 materials-14-06322-t002:** Polarization parameters for uncoated and coated samples.

Sample		Coating	E_corr_ (mV)	I_corr_ (A/cm^2^) 10^−9^	PE %
Wrought titanium	Fully dense	-	−285	14.5	-
		CS	−230	13.3	8.3
		CS-AgNPs	−112	8.4	42.3
		CS-HA	−134	12.0	17.2
PM conventional	0 vol.%	-	−142	65.0	-
		CS	−161	9.3	85.6
		CS-AgNPs	−106	2.7	95.8
		CS-HA	−123	41.5	36.1
Space-holder technique	30 vol.%	-	−164	61.7	-
		CS	−172	18.6	69.8
		CS-AgNPs	−112	9.5	84.6
		CS-HA	−112	8.8	85.8
	40 vol.%	-	−146	1330	-
		CS	−79	636	52.2
		CS-AgNPs	−67	445	66.5
		CS-HA	−54	155	88.3
	50 vol.%	-	−119	320	-
		CS	−133	19.3	93.9
		CS-AgNPs	−84	12.8	96.0
		CS-HA	−98	9.9	96.9
	60 vol.%	-	−118	1260	-
		CS	−129	306	75.7
		CS-AgNPs	−61	291	76.9
		CS-HA	−109	321	74.5

**Table 3 materials-14-06322-t003:** EIS equivalent circuit parameters obtained from EIS data of uncoated samples.

Material	R_s_ (Ω/cm^2^)	C_out_-C (µF/cm^2^)	C_out_-n	R_out_ (kΩ/cm^2^)	C_in_-C (µF/cm^2^)	C_in_-n	R_in_ (kΩ/cm^2^)	χ^2^ 10^−4^
Ti MG	23.32	27.6	0.91	158.9	3.4	0.96	379.5	13.0
Ti 0 vol.%	21.15	30.2	0.90	43.9	7.2	0.59	245.1	2.6
Ti 30 vol.%	25.12	60.6	0.81	0.06	6.6	0.95	432.2	4.4
Ti 40 vol.%	26.22	69.8	0.76	0.04	10.6	0.92	81.9	1.8
Ti 50 vol.%	17.84	49.3	0.77	0.04	13.8	0.81	131.9	6.6
Ti 60 vol.%	22.33	138.4	0.67	0.24	7.4	0.91	46.4	8.6

**Table 4 materials-14-06322-t004:** EIS equivalent circuit parameters obtained from EIS data of coated samples.

Material		Coating	C_coat_-C (µF/cm^2^)	C_coat_-n	R_coat_ (kΩ/cm^2^)	C_out_-C (µF/cm^2^)	C_out_-n	R_out_ (kΩ/cm^2^)	C_in_-C (µF/cm^2^)	C_in_-n	R_in_ (kΩ/cm^2^)	PE (%)
Wrought titanium	Fully dense	CS	7.7	0.91	3.9	2.9	0.82	10.8	1.5	0.60	485.9	-
		CS-AgNPs	6.4	0.85	10.6	2.4	0.98	53.4	0.9	0.57	541.7	11.1
		CS-HA	9.1	0.90	12.5	0.2	0.56	23.1	2.1	0.71	789.1	34.7
PM conventional	0 vol.%	CS	11.5	0.92	0.9	4.6	0.88	7.4	2.2	0.89	289.1	2.8
		CS-AgNPs	11.2	0.90	4.3	1.2	0.86	14.8	4.0	0.87	367.5	25.2
		CS-HA	7.6	0.90	3.2	6.2	0.68	21.3	11.5	0.68	292.5	8.8
Space-holder technique	30 vol.%	CS	12.7	0.87	0.9	3.0	0.85	1.5	14.2	0.78	483.0	10.9
		CS-AgNPs	11.0	0.89	8.3	1.1	0.90	6.9	12.6	0.74	548.5	23.3
		CS-HA	11.8	0.89	2.5	0.5	0.85	7.8	14.2	0.59	636.5	33.2
	40 vol.%	CS	17.2	0.86	1.0	4.4	0.73	1.7	17.0	0.71	347.9	48.1
		CS-AgNPs	13.2	0.70	2.5	1.1	0.78	5.9	14,1	0,64	605.0	70.3
		CS-HA	3.3	0.99	0.1	7.7	0.90	0.2	28.7	0.60	651.7	72.1
	50 vol.%	CS	13.7	0.87	0.3	5.4	0.76	1.9	24.0	0.76	485.7	72.9
		CS-AgNPs	12.3	0.86	1.3	1.6	0.80	2.4	15.0	0.81	684.3	80.8
		CS-HA	11.8	0.88	1.8	0.2	0.90	1.8	6.3	0.66	2038.4	93.5
	60 vol.%	CS	14.8	0.84	0.9	5.0	0.81	2.1	21.5	0.50	147.9	69.1
		CS-AgNPs	14.3	0.81	0.3	3.2	0.75	2.0	19.3	0,49	145.9	68.5
		CS-HA	87.8	0.66	0.4	0.3	0.61	2.5	63.1	0.47	32.7	-

## Data Availability

The raw/processed data required to reproduce these findings cannot be shared at this time as the data also form part of an ongoing study.

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
