# Peer review of "Improved Corrosion Behavior and Biocompatibility of Porous Titanium Samples Coated with Bioactive Chitosan-Based Nanocomposites"

_materials, 2021, doi:10.3390/ma14216322_

Round 1
Reviewer 1 Report
The paper is very interesting. The submission falls within the scope of the journal. However, authors must enrich their work with further details.
In order to evaluate better the structure of the coating, cross sectional examination is needed (before and after corrosion). For this purpose, it is better to show some corresponding cross sectional micrographs together with the plan-view ones). EDS results could be summarized into the text (Figure 2).
It would also be better if the authors have applied a characterization method, like X-Ray Diffraction (XRD) in order to characterize the microstructure of the coating (before and after corrosion).
Author Response
Response in pdf file

Reviewer 2 Report
The paper presented a work about improving the corrosion resistance and biocompatibility properties of porous titanium by chitosan based coatings. The paper is well organized and the coating obtained by electrodeposition may give a guide for the porous titanium implants against the corrosion. This manuscript meets the publication criteria of Materials. However, some questions should be addressed before publication.
- In section 3.1, the thickness of the different coatings should be provided in the paper. In addition, further analyses, for example FTIR, XRD, are necessary to clearly demonstrate the composition of the coatings.
- Page 8, paragraph 2, for the 50 vol.% and 60 vol.% samples, why do the CS coated samples have more positive OCP values than the CS-AgNPs coated samples? Authors should give a comprehensive discussion on the data presented in Figure 5.
- The OCP value of CS-HA coated 40 vol.% sample presented in Figure 5 is not consistent with that shown in Figure 4b. Please check it again.
- Conclusion is too long and it should be concise as possible.
Author Response
Response in pdf file

Reviewer 3 Report
There are major issues and the manuscript should be rejected.
Issues with the concept of the manuscript:
The usage of AgNPs is not without health risks, since the silver is not only antibacterial, but toxic due to its nonspecific mechanisms: 10.1007/s12011-019-01986-y, 10.1016/j.yrtph.2020.104690, 10.1016/j.jtemb.2021.126719, 10.1016/j.aquatox.2021.105869, 10.1016/j.enmm.2021.100485 thus the recomandation to include AgNPs in implants is not without risks and should be very carefully considered.
Chitosan is biodegradable, what is the expected lifetime of the chitosan film after implant in humans? The implant last for years, what is the purpose of a corrosion protection for a brief time? What happens with NPs after chitosan degrades? They start moving in the patient body? Seems dangerous.
The film is deposited on the external surface of the material or also inside the pores? How an external film and some NPs that do not enter into the pores/channels change the corrosion inside the bulk material?
Technical issues:
Chitosan is prepared in acetic acid 2% vol and 0.15g Hydroxyapatite powder was added into chitosan solution {that volume?} and stirred at 50⁰C for 2h to obtain the homogeneous composite solution. The hydroxyapatite is soluble in acidic solutions and the high temperature does not help the nanoparticles. Please add some justifications concerning the stability in these conditions.
How was chosen the potential scan rate 50mV/min for corrosion study? Usually much lower values are used, otherwise the results are distorted (DOI: 10.1016/j.corsci.2008.12.005).
“the Ecorr of the more porous samples has a nobler value compared to the dense samples, representing an oxide film with a higher thermodynamic stability.” => the corrosions could just take place at different potentials, Icor could be more relevant (table 2). Also, I suggest to use more anodic or more cathodic potentials instead of “more noble”
Table 2: The I corr for bare materials varies in a strange manner: similar for 0% and 30 vol. % (~65), very large for 40% and 60 % (~1300) and average for 40 % (320). It is difficult to consider these results valid and reproducible and the proposed explanations from page 11 is not convincing.
EIS although the circuit could be considered valid, numerous other circuits could be used to fit the same data (overfitting is a tendency for impedance). I am not convinced that a simple incomplete semicircle is able to differentiate between Rin and Rout.
The parameters from polarization and EIS do not seem to indicate same tendencies.
The experiement were carried out in triplicates for reproducibility, then present all courves in fig 4, error bars in fig 5. An ANOVA data interpretation for all configurations must be carried out in order to identify the significant differences.
Other aspects:
More details are required in order to able the reader to reproduce the protocols: “Chitosan-AgNPs (CS-AgNPs) coatings were prepared by dissolution of 1.5 mg/ml of CS into AgNPs solution containing acetic acid 2% vol.” => what is the volume ratio? What is the AC potential for EIS? Etc.
The abstract is not informative and must be rewritten: a long list of used techniques without results (SEM, AFM, polarization, EIS), unclear expressions: “bio composites have better adherence than chitosan coatings” => what bio composites in comparison with chitosan deposed in what conditions? Please focus on the protocols and results.
Fig 7 is not completely visible.
Some improvements of English language are necessary, e.g. “using simples one step”
Author Response
The response to reviewer in pdf file.

Round 2
Reviewer 1 Report
The submission falls within the scope of the journal. The paper is very interesting. Authors have satisfactory answers to most comments (review comments). They are detailed enough to warrant a full paper. However the authors have to make some corrections (from the first revision-reviewer comments) before the publication:
XRD: i) The type of XRD-diffractometer must been referred. ii) The type of XRD-tube must been referred. iii) The PDF cards must be referred ((Chitosan, Ti, HA- PC Powder Diffraction Files, JCPDS-ICDD). iv) Add a relevant reference about the ICDD XRD database. v)XRD diagram (not spectra).
Reviewer 2 Report
Authors have made comprehensive responses on my comments. It is accepted in present form.
Author Response
- The entire manuscript has been revised in English and minor changes have been made to the text.
Reviewer 3 Report
Concerning the reply about the AgNPs usage in medicine and toxicology: I consider the arguments as not valid and superficial, e.g. ref 5 is a biosensor for usage outside the body, 6 concerns photocatalytic degradations unrelated to medicine, 7 is about gold even if the reply indicates that is about silver, NPs from other materials (Au, Cu…) are not relevant for the discussion. The claimed antibacterial activity of silver is based on general toxic ROS that kills any cell (human or pathogen) and thus it is not ok to put toxic substance inside human implants (that will affect in equal measure the human cells and the desired bone formation).
The issue regarding the fast in vivo chitosan degradation is a conceptual one. It is pointless to put chitosan that will degrade quite fast for corrosion protection of implants that must last for many years in human body. The general idea is not valid! Even in their response the authors mention that chitosan is “biodegradable biopolymer” and “bone growth into the implant”, so what is the point of a brief corrosion protection?
The authors have no idea what happens to AgNPs after chitosan biodegradation, they migrate in the human body or stay inside the bone?
The hydroxyapatite is soluble at pH~3 that is employed in the working protocol. Room temperature included. Any brief search on internet will confirm this error.
